# Machine learning model to predict the adherence of tuberculosis patients experiencing increased levels of liver enzymes in Indonesia

**Dyah Aryani Perwitasari**[1,2]*, **Imaniar Noor Faridah**[1,2], **Haafizah Dania**[1,2], **Didik Setiawan**[3], **Triantoro Safaria**[4]

1 Faculty of Pharmacy, Universitas Ahmad Dahlan, Yogyakarta, Indonesia, 2 Mahadata, Bioinformatics and Precision Medicine in Pharmaceutical Care (MABIF Centre), Universitas Ahmad Dahlan, Yogyakarta, Indonesia, 3 Faculty of Pharmacy, Universitas Muhammadiyah Purwokerto, Purwokerto, Indonesia, 4 Faculty of Psychology, Universitas Ahmad Dahlan, Yogyakarta, Indonesia

* dyah.perwitasari@pharm.uad.ac.id

**Data Availability Statement:** The data underlying the results presented in the study are available from the e-print UAD database (https://eprints.uad.ac.id/77769/).

## Abstract

Indonesia is still the second-highest tuberculosis burden country in the world. The antituberculosis adverse drug reaction and adherence may influence the success of treatment. The objective of this study is to define the model for predicting the adherence in tuberculosis patients, based on the increased level of liver enzymes. The longitudinal study using adult tuberculosis patients treated with the first line of antituberculosis was conducted prospectively. The pregnant women and patients with complications such as gout, diabetes mellitus, liver disorder and HIV were excluded. We measured the total bilirubin, aspartate aminotransferase (AST), and alanine aminotransferase (ALT) and adherence over the 2nd, 4th, and 6th months of the treatment. We used the ORANGE Data mining as the machine learning to predict the adherence. We recruited 201 patients, whereas the male participants and less than 61 years old as the dominant participants. Around 33%, 35% and 35% tuberculosis patients experienced the increase level of bilirubine, ALT and AST, respectively. There were significant differences in ALT and AST between good and poor adherence groups, especially in the female patients. The Neural Network and Random Forests were the most suitable models to predict tuberculosis patients' adherence with good Area Under The Curve (AUC).

## Introduction

Based on the Global Tuberculosis Report (2023), Indonesia was in the two-thirds of the world's tuberculosis (TB) cases in 2022 with other countries, such as: India, China, Philippine, Pakistan, Nigeria, Bangladesh and Democratic Republic of the Congo [1]. The estimation of TB cases reached 1.060.000 cases per year, and the number of deaths due to TB reached 134.000 per year in 2023 [2]. Adherence to the treatment of TB still became a challenge in adults,

**Funding:** This work was supported by the Ministry of Education, Cultural, Research and Technology (https://dikti.kemdikbud.go.id/) through Grant Numbers 075/E5/PG.02.00.PL/2023 (12 April 2023), 0254.8/LL5-INT/AL.04/2023 (17 April 2023), and 004/WCR/LPPM UAD/IV/2023, awarded to DAP. The funders had no role in the study design, data collection, analysis, decision to publish, or preparation of the manuscript.

**Competing interests:** The authors have declared that no competing interests exist.

mostly due to the loss of follow-up. Some social factors may influence the adherence of TB patients, such as stigma, discrimination, health system factors and caregiver factors [3]. However, for the adults and late adolescences, the adverse drug reaction might became the barrier of adherence [4].

Most of the adverse drug reactions experienced by TB patients are related to the metabolism of Isoniazid (INH). INH has many polymorphisms of gene encoding the protein involved in the INH absorption, distribution, metabolism and elimination [5]. Because the complexity of adverse drug reaction mechanism in TB patients, so that education and information to the TB patients, became the important health care in primary health care [6]. The previous review found that the prevalence of adverse drug reactions elated to the first line drugs varied from 8.0% to 85%. The adverse drug reactions were gastrointestinal disturbances, serious hepatotoxicity, ototoxicity, nephrotoxicity, peripheral neuropathy and cutaneous adverse drug reaction. These adverse drug reactions may cause discontinuation of treatment or modification of the regimen [7]. The previous study defined that the incidence of liver injury reached 71.2%. However, the close monitoring and early surveillance should be conducted because the asymptomatic of liver injury reached 32.6% [7,8]. Some factors were related to the incidence of hepatotoxicity due to the antituberculosis, such as age, pregnancy, malnutrition, gender, alcoholism and comorbidities [9].

Adherence is one of the important attributes which is important to reach the effectivity of treatment. The previous study, which was explored the barrier of adherence in tuberculosis patients, found that failure to complete the treatment might results in some adverse outcome. One of the factors related to the failure of complete treatment was adverse drug reactions of antituberculosis [10]. The adverse drug reaction experienced by tuberculosis patients might be worsening their condition, and led the patients' refusal to their treatment [11]. The importance of education abut TB and TB treatment to the patients is part of the health promotion conducted in the primary health care. The TB patients' knowledge, mainly about hepatotoxicity effect of TB treatment has significant association with the patients' adherence [12].

Predicting adherence is important to give intervention to non-adherence group. Machine learning can be used accurately to predict patients; adherence involving many variables [13]. A previous study conducted on Diabetes melitus type 2 patients presented the acceptable area under the curve (AUC) for the machine learning implementation to predict the patients adherence [14].

Our study is aimed to define the model for predicting the adherence barrier in TB patients, based on the hepatotoxicity adverse drug reaction.

## Methods

We recruited TB patients in primary health care, hospital and lung hospital in a longitudinal study conducted in Yogyakarta. The inclusion criteria were: adults TB patients and receiving first line tuberculosis (Fixed Dose-Combination) treatments for 6 months. The pregnant women and patients with complications such as gout, diabetes mellitus, liver disorder and HIV were excluded. We measured participants' adherence using Medication Adherence Rating Scale (MARS). MARS questionnaire has 5 questions related to the TB medications use. It has Likert scale with 5 choices, while the non-adherence options were defined by 1 and the adherence options was defined by 5. The gradual options were defined by 2 to 4, in orderly [12]. The participants were categorized into adherence and non-adherence, using the total score of MARS questionnaire. The adherence group had total score of MARS was 25, and the non-adherence group had total score of MARS was <25. The tuberculosis patients' knowledge about tuberculosis treatment, adverse drug reaction and treatment for adverse drug reaction

was assessed using short questionnaire. The questionnaire was scored based on the correct and wrong answers. The higher score of correct answer, the better knowledge [12].

We collected laboratory results, such as haemoglobin (Hb), total bilirubin, aspartate aminotransferase (AST) and alanine aminotransferase (ALT) over the 2nd, 4th and 6th months of the treatment. We also collected other symptoms which were related to the adverse drug reaction of TB treatment, such as pruritus, nausea and vomiting. We did not use the grading criteria for the adverse drug reaction, because for the laboratory data we used the continuous data. For the patients' symptoms, it was collected based on the patients' experienced. Our study has been approved by Ethical Committee of Universitas Ahmad Dahlan, number: 012002010. We conducted the written inform consent procedures to the participant before participating in the study. We started to collect the data on 8 of March 2021 and the last data was collected on 19 January 2022. We analysed the data descriptively, and used the "ORANGE Data Mining" platform to get the model to predict tuberculosis patients' adherence based on the hepatotoxicity adverse drug reaction. We used MARS score as the dependent variable, and demographic characteristics, patients' knowledge, levels of Hb, bilirubin, AST, ALT as the independent variables. We implemented kNN, Logistic Regression, Random Forest, Naïve Bayes and Neural Network. We used leave-one out cross-validation techniques to test the performance of the models. The performance of each model is reported based on the area under the curve (AUC), accuracy (Accuration of Clarification: CA), sensitivity (Recall) and specifity (F1). AUC was used to select the best model.

The detailed structure for each model includes kNN (Number of neighbours: 5, Metric: Euclidean, Weight: Uniform), Logistic Regression (Regularization type: Lasso L1, Strength C = 0.001). Random forest (Number of trees: 10, Replicable training, No split subsets smaller than 5), Neural Network (Neurons in the hidden layers: 100,50,20, activation: ReLu, solver: Adam, Regularization α: 0.0001 and Maximum number of iterations: 200).

Separately, we also used SPSS (Version 27.0, IBM Corp) as the software with Student-T test to assess the associations between variables and adherence (significant value $< 0.05$). The methods in this study were carried out in accordance with relevant guidelines and regulations.

## Results

We recruited 201 tuberculosis patients in this study. Table 1 presents the demographic characteristics of the participants. The dominant proportion of characteristics in this study were male (58.7%), patients with age $\leq 60$ years old (85.6%), patients with normal BMI (55.7%), patients with a job (54.7%), patients with monthly income between 63.33–190.012 US Dollar (69.2%), patients with last education was up to senior high school (86.1%), and patients without comorbidity (88.1%). As much as 5% of tuberculosis patients got other medications, rather than antituberculosis. The increase of bilirubine, AST and ALT were experienced by 16.4%, 17.4% and 17.4%, respectively. There are three different regimens for antituberculosis treatment. Beside the Fixed Dose Combination (Rifampicin, Isoniazid, Pyrazinamide and Ethambutol), there were two other regiments. The two other regiments showed the 4th month treatment after the two months of initial treatment [15].

Table 2 presents the association of female tuberculosis patients and increased of liver enzymes both in adherence and non-adherence groups.

Fig 1 describes the adherence and knowledge during the 6 months of antituberculosis treatment. The tuberculosis adherences were not significantly different per month. However, the tuberculosis knowledge was getting better during the treatment.

Fig 2 presents the mean of antituberculosis adverse drug reaction experienced by participants, the average of Hb, bilirubine, ALT and AST during the 6 months of treatment. Similar

**Table 1. Socio-demographic and economic characteristics of study participants (n = 201).**

| Characteristics | Description | Frequency (%) |
|---|---|---|
| Sex | Male | 118 (58.7) |
| | Female | 83 (41.3) |
| Age | ≤ 60 years old | 172 (85.6) |
| | > 60 years old | 28 (13.9) |
| BMI Category | None | 27 (13.4) |
| | Underweight | 60 (29.9) |
| | Normal | 112 (55.7) |
| | Overweight | 2 (1.0) |
| Occupation | | |
| | Working | 110 (54.7) |
| | Not Working | 84 (41.8) |
| Monthly Income (US Dollar) | None | 139 (69.2) |
| | ≤ 63.334 | 9 (4.5) |
| | > 63.334–190.012 | 45 (22.4) |
| | > 190.012 | 8 (4.0) |
| Educational | Unknown | 8 (4.0) |
| | Up to Senior High School | 173 (86.1) |
| | Undergraduate | 19 (9.5) |
| | Postgraduate | 1 (0.5) |
| Status of Comorbidity | With Comorbidities | 24 (11.9) |
| | Without any Comorbidities | 177 (88.1) |
| Specific Comorbidity | None | 177 (88.1) |
| | Diabetes mellitus | 10 (5.0) |
| | Hypertension | 2 (1.0) |
| | Asthma | 2 (1.0) |
| | Gout | 2 (1.0) |
| | Chronic Obstructive Pulmonary Disease | 1 (0.5) |
| | Epilepsy | 1 (0.5) |
| | Others | 6 (3.0) |
| TB medication | None | 6 (3.0) |
| | Fixed-Dose Combination | 85 (42.3) |
| | Rifampicin-Isoniazide | 107 (53.2) |
| | Rifampicin-Isoniazide-Ethambutol | 2 (1.0) |
| | Rifampicin-Ethambutol | 1 (0.5) |
| Other Regular medication | With regular medication | 10 (5.0) |
| Regular medication details | Without regular medication | 191 (95.0) |
| Patient experienced decrease of Hb | None | 191 (95.0) |
| | Antiallergy | 1 (0.5) |
| | Antihyperuricemia | 2 (1.0) |
| | Antihypertension | 1 (0.5) |
| | Cough medicine | 3 (1.5) |
| | Bronchodilators | 2 (1) |
| | Multivitamin | 1 (0.5) |
| | No | 178 (88.6) |
| | Yes | 23 (11.4) |
| Patient experienced increased of bilirubin | No | 168 (83.6) |
| | Yes | 33 (16.4) |
| Patient experienced increased of AST | No | 166 (82.6) |
| | Yes | 35 (17.4) |
| Patient experience increased of ALT | No | 166 (82.6) |
| | Yes | 35 (17.4) |

AST: Aspartate Aminotransferase; ALT: Alanine Transaminase.

**Table 2. The results of the association test between laboratory data and TB patients' adherence in the 2nd, 4th and 6th months.**

| Laboratory parameters | 2nd month | | | 4th month | | | 6th month | | |
|---|---|---|---|---|---|---|---|---|---|
| Mean ± SD | Adherence | Non-adherence | P value | Adherence | Non-adherence | P value | Adherence | Non-adherence | P value |
| Hb (g/dL)<br>F: 12–14<br>M: 14–16 | 13.5 ± 1.93<br>13.12 ± 1.94<br>13.9 ± 1.82 | 13.6±1.72<br>13.7 ±0.84<br>14.4 ±. 2.77 | 0.888<br>0.683<br>0.651 | 13.6± 2.04<br>13.4 ± 1.92<br>13.4 ± 2.11 | 13.4±2.03<br>13.7 ± 0.84<br>14.4 ± 2.77 | 0.736<br>0.806<br>0.754 | 13.8+1.82<br>13.3 ± 1.90<br>14.28 ± 1.67 | 14.1+2.04<br>13.7 ± 0.84<br>14.42 ± 2.77 | 0.750<br>0.789<br>0.938 |
| Bilirubin (mg/dL)<br>0.25–1 | 1.4±2.86 | 2.0±4.09 | 0.545 | 1.3+2.85 | 2.0+3.84 | 0.487 | 1.4+3.06 | 3.1+5.61 | 0.246 |
| ALT (U/L)<br>F <23<br>M <30 | 22.1±16.32<br>22.9 ± 9.3<br>25.9 ± 11.90 | 26.5±25.19<br>39.5 ± 24.74<br>39.4 ± 12.96 | 0.489<br>0.024*<br>0.062 | 21.2+13.30<br>22.3 ± 9.09<br>24.5 ± 10.94 | 25.8+27.46<br>39.5 ± 24.74<br>39.4 ± 12.96 | 0.384<br>0.019*<br>0.02* | 21.5+13.91<br>22.2 ± 9.09<br>25.5 ± 11.55 | 35.3+31.75<br>39.5 ± 24.74<br>39.4 ± 12.96 | 0.045*<br>0.016*<br>0.04* |
| AST (U/L)<br>F <21<br>M <25 | 24.9±11.00<br>18.7 ± 13.55<br>24.8 ± 17.52 | 28.9±16.34<br>50.5 ± 55.86<br>25.20 ± 8.51 | 0.323<br>0.007*<br>0.975 | 23.8+10.25<br>18.7 ± 13.73<br>22.65 ± 12.72 | 29.5+18.00<br>50.5 ± 55.86<br>25.2 ± 8.51 | 0.155<br>0.007*<br>0.734 | 24.0+10.63<br>18.8 ± 13.00<br>23.6 ± 14.35 | 39.4+15.39<br>50.5 ± 55.86<br>25.2 ± 8.51 | 0.002*<br>0.006*<br>0.850 |

Hb: Hemoglobin; F: Female; M: Male; mg/dL: milligram/deciliter; ALT: Alanine Transaminase; AST: Aspartate Aminotransferase; U/L: unit/liter.

pattern can be seen in the variables in Fig 2. At the end of the treatment, the means of the parameters increased.

Fig 3 presents the flowchart of ORANGE Data Mining.

Table 3 presents the results of machine learning analysis using some models, such as kNN, Random Forest, Neural Network, Naïve Bayes, and Logistic Regression. The good model can be seen in Random Forest, Neural Network and Naïve Bayes with area under the curve (AUC) value more than 80%. However, the accuration of clarification (CA), F score (Ft) and Recall values of Naïve Bayes were not good.

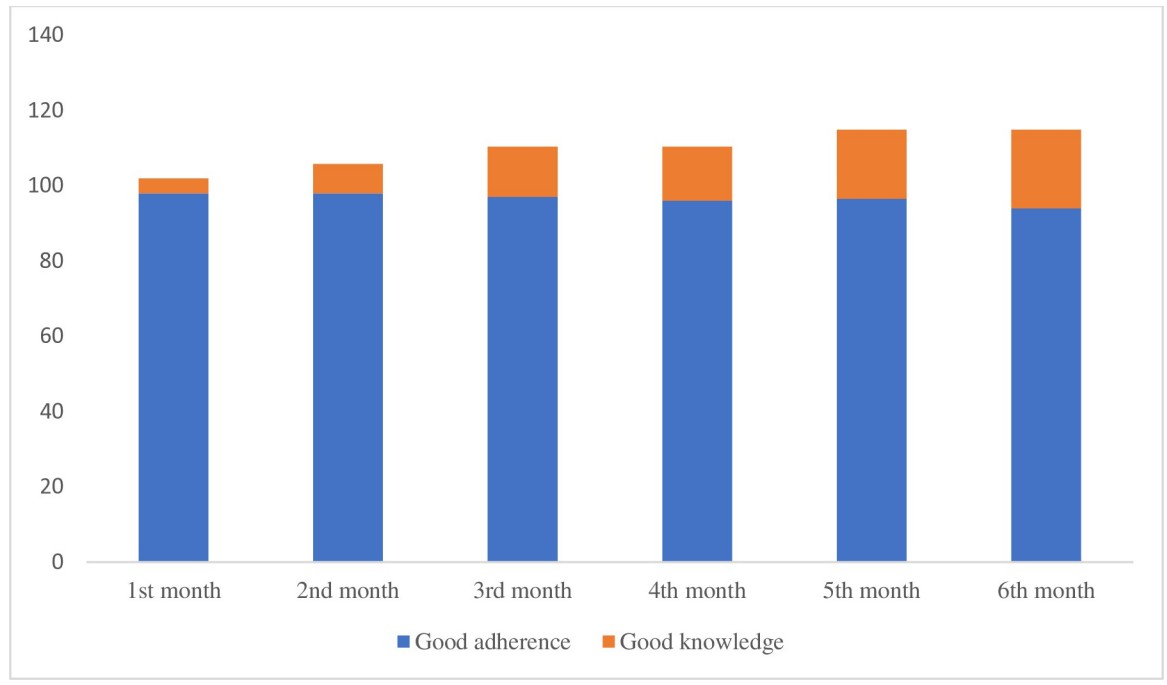

**Fig 1.**

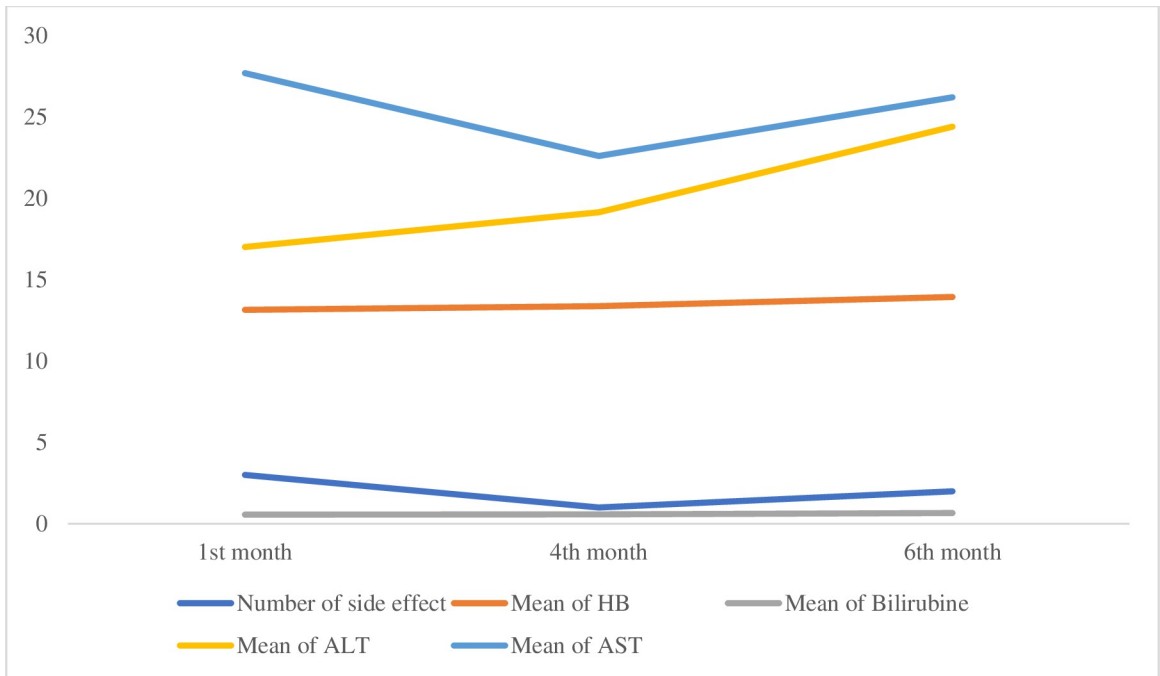

**Fig 2.**

Fig 4 presents the results of ROC analysis among the models. The good AUC can be seen in the Neural Network and Random Forest models.

## Discussion

Our study finds that tuberculosis patients had a good adherence and knowledge, during the 6 months of treatment. Some of them, experienced the increased of liver enzymes at the end of

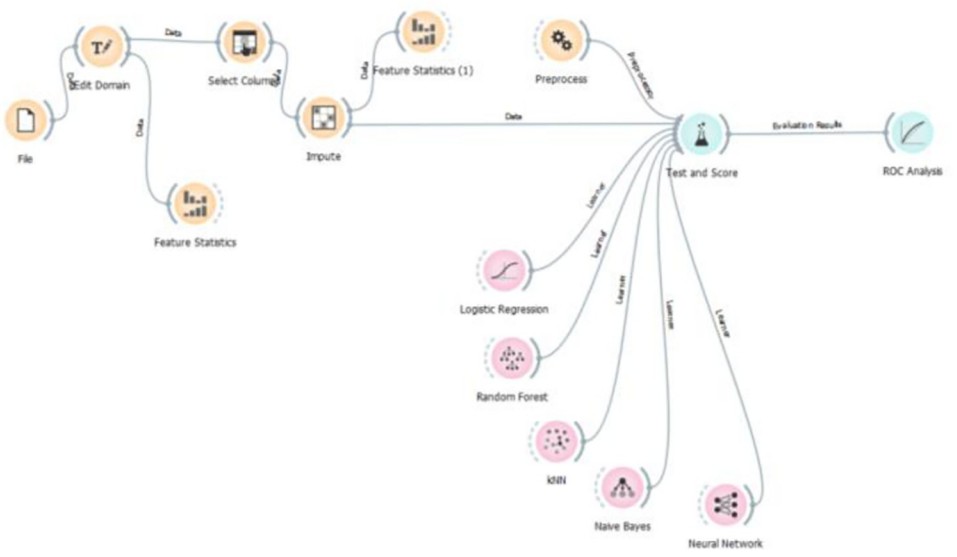

**Fig 3.**

**Table 3. Models of adherence prediction in tuberculosis patients using ORANGE data mining.**

| Model | AUC | CA | F1 | Precision | Recall |
|---|---|---|---|---|---|
| kNN | 0.634 | 0.911 | 0.911 | 0.884 | 0.940 |
| Random Forest | 0.808 | 0.954 | 0.954 | 0.956 | 0.960 |
| Neural network | 0.801 | 0.952 | 0.952 | 0.951 | 0.955 |
| Naïve Bayes | 0.891 | 0.599 | 0.599 | 0.947 | 0.488 |
| Logistic regression | 0.5 | 0.911 | 0.911 | 0.884 | 0.940 |

AUC: Area Under the Curve; CA: Accuration of Clarification; F1: F-score.

the treatment. Adherence prediction is very important to reach the treatment effectiveness. Many factors may influence the adherence of tuberculosis patients in taking the medicine. Our study also finds that adherence can be predicted by a model including the characteristics demographic and also the level of liver enzymes. The most appropriate model for this prediction is Random Forest and Neural Network.

The good adherence of tuberculosis patients can be influenced by some factors. A systematic review in Indian subcontinent presents that tuberculosis patients adherence can be influenced by various factors which were correlated with the treatment. However, the role of the health providers, including the tuberculosis programmers [16]. The Ministry of Health in Indonesia, also decided that Direct Observed Treatment (DOT) for tuberculosis patients included in the strategy of tuberculosis elimination program in Indonesia [17]. DOTs can support the tuberculosis patients, especially in avoiding the taking of antituberculosis. The role of DOTs in increasing tuberculosis patients' adherence is significant [18]. In the National Strategy for tuberculosis elimination in Indonesia, the government support the DOT's role, such as the salary for nurse who can be the DOTs of tuberculosis patients, the rule that DOTs for tuberculosis patients must be from the family or closest neighbor [17]. All of the tuberculosis patients in this study were supported by DOTs. Thus, the tuberculosis patients' adherence and knowledge about hepatotoxicity adverse drug reaction are good. The previous study, also

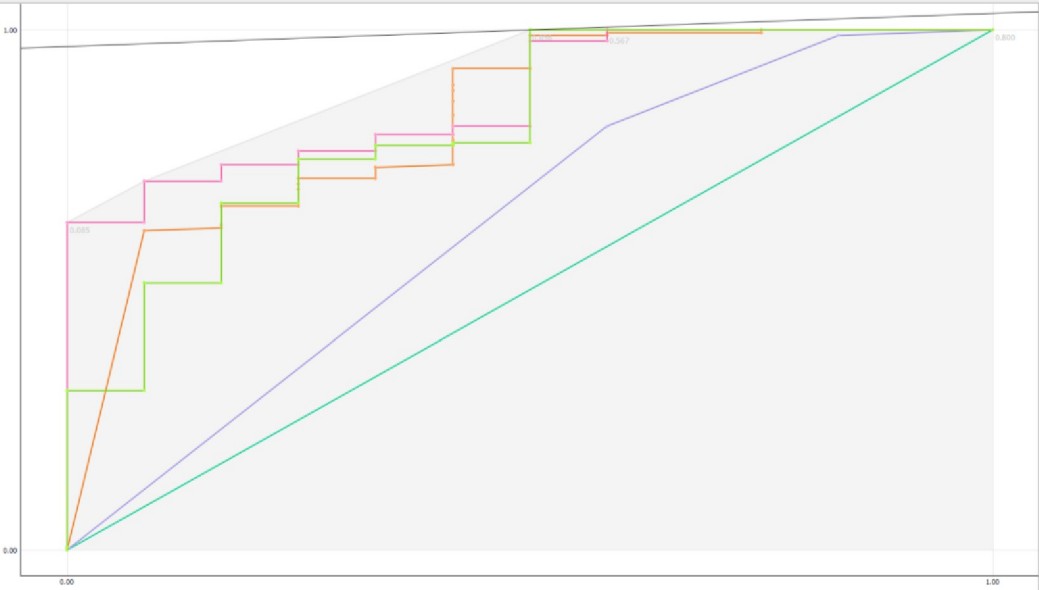

**Fig 4. ROC of Naïve Bayes, Random Forest, Neural Network, kNN and logistic regression.**

conducted in Indonesia, described the significant association between tuberculosis patients' adherence and the hepatotoxicity due to the antituberculosis treatment [12]. Female tuberculosis patients have a significant association with increased of liver enzymes, in both of adherence and non-adherence groups. Our study is similar to a previous study in Taiwan [19]. The female factor predicts the hepatotoxicity adverse drug reaction with other factors, such as old age, immunocompromised disease and duration of treatment.

Using the machine learning to predict adherence in tuberculosis is still rare. However, in other diseases, the use of machine learning in medication adherence prediction can be found in many studies. The previous scoping review about the use of machine learning in predicting medication adherence mentioned that machine learning could be used to predict medication adherence with a good level of accuracy [20]. Some predictors was found during the machine learning analysis, duration level, marital status, income, gender, geographic location, emergency care interventions, age, race, ethnicity, disease severity, comorbidities, medication cost, insurance coverage, substance abuse, medication beliefs, medication knowledge, medication dose, medication frequency, initial medication adherence, and current medications [21–23].

Our study finding is online with previous study in adherence prediction. The Neural Networks model is appropriate for adherence prediction [23]. The Neural Network and Random Forest can be used as the model to predict medication adherence, which involving many variables. Our study has limitation that the sample size, because the research conducted during the pandemic era. The baseline parameters of the tuberculosis patients were not collected in this study because we implemented particular inclusion and exclusion criteria and we did the prospective monitoring in this study. However, the strength of this study, was related to the use of machine learning to predict the medication adherence in Indonesia. Also, we followed the patients' treatment until 6 months, thus we can do the monitoring of adverse drug reaction and liver functions during the treatment. One of the attributes implemented in machine learning are level of liver enzymes and the elevation of the liver enzymes. There are some patients experiencing the elevation of liver enzymes at the end of the treatment. The tuberculosis patients' knowledge also defined in this study.

The implementation of this study in the clinical practice is to involve the education about hepatotoxicity after antituberculosis treatment, laboratory parameters and characteristic demography as the variables to increase the tuberculosis patients' adherence. Future studies are suggested to include more attributes related to the psycho-behaviour factors.

## Conclusion

This study developed the model for predicting the tuberculosis patients adherence. The Neural Network model was the most suitable model to predict the adherence of tuberculosis patients based on some variables related to the increased of liver enzymes during the treatment.

## Acknowledgments

The authors thank to the local government for the study permition.

## Author Contributions

**Conceptualization:** Dyah Aryani Perwitasari.

**Data curation:** Haafizah Dania, Triantoro Safaria.

**Formal analysis:** Haafizah Dania.

**Funding acquisition:** Dyah Aryani Perwitasari.

**Investigation:** Imaniar Noor Faridah.

**Methodology:** Haafizah Dania, Triantoro Safaria.

**Project administration:** Imaniar Noor Faridah.

**Resources:** Didik Setiawan.

**Software:** Didik Setiawan.

**Validation:** Dyah Aryani Perwitasari, Triantoro Safaria.

**Visualization:** Didik Setiawan.

**Writing – original draft:** Dyah Aryani Perwitasari.

**Writing – review & editing:** Dyah Aryani Perwitasari, Imaniar Noor Faridah, Triantoro Safaria.

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
