## [Decision Letter · Decision Letter 0]

6 Nov 2024

PONE-D-24-35376Machine Learning Model to Predict The Adherence of Tuberculosis Patients Experiencing Increased Level of Liver Enzymes in IndonesiaPLOS ONE

Dear Dr. Perwitasari,

Thank you for submitting your manuscript to PLOS ONE. After careful consideration, we feel that it has merit but does not fully meet PLOS ONE’s publication criteria as it currently stands. Therefore, we invite you to submit a revised version of the manuscript that addresses the points raised during the review process.

Please submit your revised manuscript by Dec 21 2024 11:59PM. If you will need significantly more time to complete your revisions, please reply to this message or contact the journal office at plosone@plos.org. Please include the following items when submitting your revised manuscript:A rebuttal letter that responds to each point raised by the academic editor and reviewer(s). You should upload this letter as a separate file labeled 'Response to Reviewers'.A marked-up copy of your manuscript that highlights changes made to the original version. You should upload this as a separate file labeled 'Revised Manuscript with Track Changes'.An unmarked version of your revised paper without tracked changes. You should upload this as a separate file labeled 'Manuscript'.

We look forward to receiving your revised manuscript.

Kind regards,

Frederick Quinn

Academic Editor

PLOS ONE

**Journal Requirements:**

The author thank to Ministry of Education, Cultural, Research and Technology for the grant research Number 075/E5/PG.02.00.PL/2023 date12 April 2023; Number : 0254.8/LL5-INT/AL.04/2023 date 17 April 2023; and Number 004/WCR/LPPM UAD/IV/2023. 

Author who received award: DAP

Funder: Ministry of Education, Cultural, Research and Technology for the grant research Number 075/E5/PG.02.00.PL/2023 date12 April 2023; Number : 0254.8/LL5-INT/AL.04/2023 date 17 April 2023; and Number 004/WCR/LPPM UAD/IV/2023. 

URL: https://dikti.kemdikbud.go.id/

The funders did not play any role in the study design, data collection, analysis, decision to publish orr preparation of manuscript

Reviewers' comments:

Reviewer's Responses to Questions

**Comments to the Author**

1. Is the manuscript technically sound, and do the data support the conclusions?

Reviewer #1: No

2. Has the statistical analysis been performed appropriately and rigorously? 

Reviewer #1: No

3. Have the authors made all data underlying the findings in their manuscript fully available?

Reviewer #1: Yes

4. Is the manuscript presented in an intelligible fashion and written in standard English?

Reviewer #1: Yes

5. Review Comments to the Author

**Reviewer #1:** Dear authors,

Thank you for submitting your manuscript titled "Machine Learning Model to Predict the Adherence of Tuberculosis Patients Experiencing Increased Levels of Liver Enzymes in Indonesia." This is an important and timely topic with potential clinical implications. However, there are several areas that need further clarification and improvement, particularly in terms of language, methodology, and the presentation of results, to ensure the study's findings are clearly understood and supported, as follow below:

Abstract

• Needs to be re-written after the manuscript review

Introduction

• English review is needed

• Suggest to change the wording of “side effect” to “adverse drug reaction”

• Contextualize more the hepatic adverse drug reaction

• Revise this sentence: “The poor adherence may cause the adverse drug reaction (…)” – in fact, is the opposite.

• Rewrite the sentence: “The side effect experienced by the TB patients led the patients refused the treatment. This side effect might be worsening TB patients condition, and led the patients’ refusal to their treatment”

• Overall: needs improvement, English review, and better paragraphs linkage

Methods

• Where the study was conducted?

• Clarify the inclusion criteria – what clinical forms were considered? Which anti-TB drugs or anti-TB drugs combination defined the first line? Was it a fixed-dose combination?

• Define which were the “comorbidities which related to liver function were excluded”

• Question: why did the authors exclude the pregnant women? What was the age range for inclusion?

• Define this better: “We also collected other symptoms which were related to the side effect of TB treatment.” The adverse drug reaction symptoms are usually related to the organ system affected. Which organ system(s) considered? Only hepatic?

• It’s not clear which variables were included in the model and how they were selected.

• Review this paragraph “Separately, we also used SPSS (Version 27.0, IBM Corp) to assess the associations between variables and adherence. The methods in this study were carried out in accordance with relevant guidelines and regulations.”

• SPSS is the statistical software. What were the statistical tests used for the associations? Why? What was alfa value, .05?

Results

• Review and rewrite the first paragraph – it’s confusing and difficult to follow.

• Table 1 – please provide a footnote with abbreviation

• Question: what is IDR? Not previously defined. I suppose it’s the Indonesian currency, and I suggest to also provide the equivalency in dollars or euros, so the reader can have a better perception of the amount.

• The variables described are not previously defined – e.g., how level of education was evaluated?

• There is no descriptive statistics of the baseline variables, of the adverse drug reaction characteristics, median AST, ALT, Hb…

• There is a variable “TB medication” in table 1, which list different categories of TB treatment. Please define and clarify.

• Moreover, there is a category of “regular medication details”. I suggest grouping them by medication class, such as anti-hypertension, medication for diabetes, and so on.

• Table 2 is not informative, lacks a footnote with abbreviation, and the statistical test used to compare. E.g., what is F<21?

• Figure 1 – again, where are the statistical results? Which test was used to compare de proportions of non-adherence?

Discussion

• Review this sentence:

• “Our study finds that tuberculosis patients had a good adherence and knowledge” – there is no mention of how “knowledge” was handled.

• Review the other paragraphs according to the suggestions in methods and results sections

• Conclusion

• Invalid for now. The study did not assess the implementation of a prediction model, but a development of it.

General comment:

• Please avoid using “subjects” when referring to study participants. Prefer participants, volunteers, or study participants.

Final comment:

In summary, while your study addresses the important issue of adverse drug reaction during TB treatment, several areas require further attention before it can be considered for publication. The manuscript would benefit from a thorough review of English language and structure, clearer definitions of key variables, improved descriptions of the methods and statistical analyses, and more robust presentation of the results. Additionally, the discussion and conclusion sections need to be better aligned with the findings. I recommend a major revision to address these points and strengthen the overall clarity and rigor of the study.

6. PLOS authors have the option to publish the peer review history of their article (what does this mean?). If published, this will include your full peer review and any attached files.

Reviewer #1: **Yes: **Felipe Ridolfi

---

## [Author Response · Author response to Decision Letter 0]

14 Nov 2024

Dear Editor,

Thank you for the response ang also constructive comments and suggestions for our manuscript (PONE-D-24-35376), entitled: Machine Learning Model to Predict the Adherence of Tuberculosis Patients Experiencing Increased Level of Liver Enzymes in Indonesia.

We revised the manuscript based on the Editor/Reviewer comments. We put the revision in the rebuttal letter, also in the manuscript with the track changes. We also attach the proof for English editing.

We look forward to hearing from you regarding our submission. We would be glad to respond to any further questions and comments that you may have

Corresponding author,

Dyah Aryani Perwitasari

---

## [Decision Letter · Decision Letter 1]

26 Nov 2024

PONE-D-24-35376R1Machine Learning Model to Predict The Adherence of Tuberculosis Patients Experiencing Increased Level of Liver Enzymes in IndonesiaPLOS ONE

Dear Dr. Perwitasari,

Thank you for submitting your manuscript to PLOS ONE. After careful consideration, we feel that it has merit but does not fully meet PLOS ONE’s publication criteria as it currently stands. Therefore, we invite you to submit a revised version of the manuscript that addresses the points raised during the review process.

Please submit your revised manuscript by Jan 10 2025 11:59PM. If you will need significantly more time to complete your revisions, please reply to this message or contact the journal office at plosone@plos.org. Please include the following items when submitting your revised manuscript:A rebuttal letter that responds to each point raised by the academic editor and reviewer(s). You should upload this letter as a separate file labeled 'Response to Reviewers'.A marked-up copy of your manuscript that highlights changes made to the original version. You should upload this as a separate file labeled 'Revised Manuscript with Track Changes'.An unmarked version of your revised paper without tracked changes. You should upload this as a separate file labeled 'Manuscript'.If applicable, we recommend that you deposit your laboratory protocols in protocols.io to enhance the reproducibility of your results. Protocols.io assigns your protocol its own identifier (DOI) so that it can be cited independently in the future. For instructions see: https://journals.plos.org/plosone/s/submission-guidelines#loc-laboratory-protocols. Additionally, PLOS ONE offers an option for publishing peer-reviewed Lab Protocol articles, which describe protocols hosted on protocols.io. Read more information on sharing protocols at https://plos.org/protocols?utm_medium=editorial-email&utm_source=authorletters&utm_campaign=protocols.

We look forward to receiving your revised manuscript.

Kind regards,

Frederick Quinn

Academic Editor

PLOS ONE

Journal Requirements:

Reviewers' comments:

Reviewer's Responses to Questions

**Comments to the Author**

1. If the authors have adequately addressed your comments raised in a previous round of review and you feel that this manuscript is now acceptable for publication, you may indicate that here to bypass the “Comments to the Author” section, enter your conflict of interest statement in the “Confidential to Editor” section, and submit your "Accept" recommendation.

Reviewer #1: All comments have been addressed

2. Is the manuscript technically sound, and do the data support the conclusions?

Reviewer #1: Partly

3. Has the statistical analysis been performed appropriately and rigorously? 

Reviewer #1: Yes

4. Have the authors made all data underlying the findings in their manuscript fully available?

Reviewer #1: Yes

5. Is the manuscript presented in an intelligible fashion and written in standard English?

Reviewer #1: Yes

6. Review Comments to the Author

Reviewer #1: Dear authors, thanks for sending back your revised manuscritp. It has improved after your revision. However, I believe it would benefit from one last very careful review of grammar and wording, as there are some repeated small inconsistences - e.g., in abstract, you mention "adverse reaction", while in text, "adverse drug reaction". Moreover, the letter style in the figures differ from the text.

And just one question: I did not see any grading criteria for the adverse drug reaction - if you did not use any, please specify in methods.

7. PLOS authors have the option to publish the peer review history of their article (what does this mean?). If published, this will include your full peer review and any attached files.

Reviewer #1: **Yes: **Felipe Ridolfi

---

## [Author Response · Author response to Decision Letter 1]

2 Dec 2024

Dear Editor

Thank you for the review after we revised our manuscript (PONE-D-24-35376R1).

Below is our response to the reviewer suggestion. 

We look forward to hearing from you regarding our submission. We would be glad to respond to any further questions and comments that you may have

Best regards,

Corresponding letter

Dyah Aryani Perwitasari 

Response to the editor and reviewer

Journal Requirements:

Response

1. The reference list is based on the cited reference in the body of text

2. We do not have any retracted publication

Comments to the Author

1. If the authors have adequately addressed your comments raised in a previous round of review and you feel that this manuscript is now acceptable for publication, you may indicate that here to bypass the “Comments to the Author” section, enter your conflict of interest statement in the “Confidential to Editor” section, and submit your "Accept" recommendation.

Reviewer #1: All comments have been addressed

2. Is the manuscript technically sound, and do the data support the conclusions?

Reviewer #1: Partly

3. Has the statistical analysis been performed appropriately and rigorously?

Reviewer #1: Yes

4. Have the authors made all data underlying the findings in their manuscript fully available?

Reviewer #1: Yes

5. Is the manuscript presented in an intelligible fashion and written in standard English?

Reviewer #1: Yes

Response : Thank you for accepting my revisions

6. Review Comments to the Author

Reviewer #1: Dear authors, thanks for sending back your revised manuscritp. It has improved after your revision. However, I believe it would benefit from one last very careful review of grammar and wording, as there are some repeated small inconsistences - e.g., in abstract, you mention "adverse reaction", while in text, "adverse drug reaction". Moreover, the letter style in the figures differ from the text.

And just one question: I did not see any grading criteria for the adverse drug reaction - if you did not use any, please specify in methods.

Response:

Thank you for the comments and suggestion.

We already revised the adverse reaction into adverse drug reaction.

We already adjust the letter style in the figures.

We did not use the grading criteria, and put this statement in the method : 

We did not use the grading criteria for the adverse drug reaction, because for the laboratory data we used the continuous data. For the patients’ symptoms, it was collected based on the patients’ experienced.

Response :

We already proceed our figures in PACE and upload separated in the system

---

## [Decision Letter · Decision Letter 2]

4 Dec 2024

Machine Learning Model to Predict The Adherence of Tuberculosis Patients Experiencing Increased Level of Liver Enzymes in Indonesia

PONE-D-24-35376R2

Dear Dr. Perwitasari,

We’re pleased to inform you that your manuscript has been judged scientifically suitable for publication and will be formally accepted for publication once it meets all outstanding technical requirements.

Kind regards,

Frederick Quinn

Academic Editor

PLOS ONE

Additional Editor Comments (optional):

Reviewers' comments:

Reviewer's Responses to Questions

**Comments to the Author**

1. If the authors have adequately addressed your comments raised in a previous round of review and you feel that this manuscript is now acceptable for publication, you may indicate that here to bypass the “Comments to the Author” section, enter your conflict of interest statement in the “Confidential to Editor” section, and submit your "Accept" recommendation.

Reviewer #1: All comments have been addressed

2. Is the manuscript technically sound, and do the data support the conclusions?

Reviewer #1: Yes

3. Has the statistical analysis been performed appropriately and rigorously? 

Reviewer #1: Yes

4. Have the authors made all data underlying the findings in their manuscript fully available?

Reviewer #1: Yes

5. Is the manuscript presented in an intelligible fashion and written in standard English?

Reviewer #1: Yes

6. Review Comments to the Author

Reviewer #1: (No Response)

7. PLOS authors have the option to publish the peer review history of their article (what does this mean?). If published, this will include your full peer review and any attached files.

Reviewer #1: **Yes: **Felipe Ridolfi

---

## [Editor Report · Acceptance letter]

17 Dec 2024

PONE-D-24-35376R2 

PLOS ONE

Dear Dr. Perwitasari, 

I'm pleased to inform you that your manuscript has been deemed suitable for publication in PLOS ONE. Congratulations! Your manuscript is now being handed over to our production team.

Kind regards, 

on behalf of

Dr. Frederick Quinn 

Academic Editor

PLOS ONE